# Fast and Sensitive Detection of Soil-Borne Cereal Mosaic Virus in Leaf Crude Extract of Durum Wheat

**DOI:** 10.3390/v15010140

**Published:** 2022-12-31

**Authors:** Monica Marra, Chiara D’Errico, Cinzia Montemurro, Claudio Ratti, Elena Baldoni, Slavica Matic, Gian Paolo Accotto

**Affiliations:** 1Institute for Sustainable Plant Protection, National Research Council, 10135 Turin, Italy; 2Department of Soil, Plant and Food Science, University of Bari Aldo Moro, 70126 Bari, Italy; 3European Laboratory for Non-Linear Spectroscopy, LENS, 50019 Sesto Fiorentino, Italy; 4Institute for Sustainable Plant Protection, National Research Council, 70126 Bari, Italy; 5Department of Agricultural and Food Sciences, University of Bologna, 40127 Bologna, Italy; 6Institute of Agricultural Biology and Biotechnology, National Research Council, 20133 Milan, Italy

**Keywords:** SBCMV, durum wheat, RT-LAMP, real-time PCR, *Furovirus*

## Abstract

Soil-borne cereal mosaic virus (SBCMV) is a furovirus with rigid rod-shaped particles containing an ssRNA genome, transmitted by *Polymyxa graminis* Led., a plasmodiophorid that can persist in soil for up to 20 years. SBCMV was reported on common and durum wheat and it can cause yield losses of up to 70%. Detection protocols currently available are costly and time-consuming (real-time PCR) or have limited sensitivity (ELISA). To facilitate an efficient investigation of the real dispersal of SBCMV, it is necessary to develop a new detection tool with the following characteristics: no extraction steps, very fast results, and high sensitivity to allow pooling of a large number of samples. In the present work, we have developed a reverse transcription loop-mediated isothermal amplification (RT-LAMP) protocol with such characteristics, and we have compared it with real-time PCR. Our results show that the sensitivity of LAMP and real-time PCR on cDNA and RT-LAMP on crude extracts are comparable, with the obvious advantage that RT-LAMP produces results in minutes rather than hours. This paves the way for extensive field surveys, leading to a better knowledge of the impact of this virus on wheat health and yield.

## 1. Introduction

Durum wheat (*Triticum turgidum* L. subsp. *durum* (Desf.) Husn.) contributes 14% of total EU wheat production, and Italy represents the greatest European producer [1]. Due to its technological properties, durum wheat is used to produce the core business “pasta” and some typical breads [2]. This cereal may be affected by numerous plant pathogens, among which several viruses. In particular, soil-borne wheat viruses have been reported throughout the world [3], but the damage they cause is very difficult to evaluate, due to similar symptomatology shared with few physiological stresses. Their spread and severity are also predicted to be increased by climatic change [4,5].

In Europe, the most-known soil-borne virus infecting wheat is soil-borne cereal mosaic virus (SBCMV), a furovirus with rigid rod-shaped particles containing an ssRNA genome [6]. SBCMV is transmitted by *Polymyxa graminis* Led., a plasmodiophorid that can persist in soil for up to 20 years [7,8]. In addition to durum wheat, SBCMV was reported on common wheat (*Triticum aestivum* L.), as well as on the less cultivated rye and the hybrid cereal triticale [3]. Information about the actual spread of this virus is still limited, but recently it gained increased interest in Europe due to yield losses of up to 70% in infected wheat fields [9,10,11,12]. The symptoms of the virus are visible in late winter or early spring and then tend to disappear as temperatures rise. Infected plants show a general yellowing and elongated chlorotic spots on leaves, along the veins. The infection can also reduce the kernel weight and the plant height [13,14].

Another virus reported in Europe, the bymovirus wheat spindle streak mosaic virus (WSSMV), is often found associated with SBCMV [15]. The symptoms caused by wheat bymoviruses are similar to the symptoms caused by furoviruses and both virus genera are transmitted by *P. graminis* [16,17]. When plants are infected by SBCMV and WSSMV, grain yields can be reduced by 50–70% [18]. The third soil-borne virus described in some parts of Central Europe is soil-borne wheat mosaic virus (SBWMV), a furovirus closely related to SBCMV [19].

There are no direct and effective control methods against the disease caused by SBCMV, SBWMV, or WSSMV. The disease is mainly managed by growing virus-resistant genotypes or by selecting *P. graminis*-free soils. Efficient detection of the virus relies on the use of sensitive and specific diagnostic methods. Currently, molecular diagnostics for SBCMV in infected wheat plants is based on expensive methods such as conventional PCR [20,21], and Taqman real-time PCR [22], while serological diagnostic techniques such as the enzyme-linked immunosorbent assay (ELISA) offer limited sensitivity [23].

Among molecular techniques, loop-mediated isothermal amplification (LAMP) assay is increasingly used in plant pathology thanks to its fast performance and the ability to be applied directly in the field [24,25]. Furthermore, its sensitivity is higher or similar to that of other molecular techniques, allowing the detection of small quantities of the target viral genome (picograms to femtograms), depending on the host/virus combination [26,27,28,29]. In addition, the high resistance of the LAMP Bst DNA polymerase enzyme to reaction inhibitors [30,31] makes it possible to skip the RNA or DNA extraction step and use directly the crude plant extracts in which tissues are mechanically disrupted and homogenised in a buffer [32,33].

Although the LAMP technique is widely applied in the quick molecular detection of numerous plant viruses [25], no isothermal amplification assay has been developed so far for the detection of SBCMV. To overcome this limitation and to allow future large-scale studies about the dispersal and severity of SBCMV, we have developed a LAMP procedure for specific and quick virus detection. Another goal of this study was to develop a rapid plant extraction procedure, avoiding the canonical nucleic acid extraction protocols and allowing a direct in-field application.

## 2. Materials and Methods

### 2.1. Plant Samples

Leaves of durum wheat infected by SBCMV were collected from an experimental station at the University of Bologna, Italy, where both the vector and the virus are present [22]. These leaves were used to propagate the virus in durum wheat plants (cv. Claudio), through mechanical inoculation. Leaves were ground in the presence of carborundum and K-phosphate buffer 50 mM pH 7, containing 1 mM sodium ethylenediaminetetraacetic acid (EDTA), 5 mM sodium diethyldithiocarbamate trihydrate (Na-DIECA), and 5 mM sodium thioglycolate. Seedlings were inoculated 10–12 days after sowing (BBCH scale 13) then maintained in a growth chamber at 19/17 °C (16 h day/8 h night). To optimise the protocol, leaves of plants grown in screenhouse in SBCMV-infected soil, taken from the abovementioned experimental field, were also used.

### 2.2. Sample Preparation

#### 2.2.1. RNA Extraction and cDNA Synthesis

Total RNA was extracted from 100 mg durum wheat leaves with Spectrum™ Plant Total RNA Kit (Sigma-Aldrich Pty. Ltd., Sydney, NSW, Australia). The extracted RNA was resuspended in 50 µL of nuclease-free water, then 10 µL were used for cDNA synthesis with the High-Capacity cDNA Reverse Transcription Kit (Applied Biosystems, Cheshire, UK) in a final volume of 20 µL.

#### 2.2.2. Crude Extract Preparation

To assess the best preparation of the crude extract, durum wheat leaves were ground in three different buffers: (i) ELISA buffer (phosphate-buffered saline pH 7.4, 10 mM potassium phosphate, 150 mM sodium chloride, Tween-20 at 5 mL/L, and polyvinyl pyrrolidone at 20 g/L); (ii) TET buffer (1% Triton X-100, 20 mM Tris-HCl pH 8, 20 mM EDTA) [34]; (iii) TRIS buffer (100 mM Tris–HCl, pH 8.0) [35]. One hundred milligrams of leaves were ground with 1 mL of buffer in Bioreba extraction bags (Bioreba, Switzerland) then the crude extracts were diluted in sterile water at dilutions from 10^−3^ to 10^−5^.

### 2.3. SBCMV Detection

#### 2.3.1. LAMP Primer Design

Two sets of primers were designed (Table 1) using the Eiken Primer Explorer software (Eiken Chemical Co., Ltd., Tokyo, Japan), based on the SBCMV (GenBank AJ132577.1) genomic region encoding the coat protein. We then selected those primers that match all the available SBCMV sequences.

The positions of the primers on the SBCMV sequence are shown in Figure 1. Primers were diluted at 50 μM each.

#### 2.3.2. LAMP and Reverse Transcription LAMP (RT-LAMP)

As a template, either 1 µL of cDNA or 2 µL of crude extract were used. Reactions were performed using the Isothermal Master Mix ISO-004^®^ (OptiGene, Horsham, UK), in a final volume of 10 µL with primers at the concentration indicated in Table 1. 

LAMP assay was tested within a range of 60–65 °C (increasing 1 °C per each test) for 30 min. Melting curves for LAMP assays were generated from 95 °C to 75 °C (ramp speed 0.05 °C/s, with plate readings every 10 s). The analyses were performed using two instruments, the CFX96 Real-time PCR Detection System (Bio-Rad, Hercules, CA, USA) or the Hyris bCUBE (Hyris, London, UK), which is a portable device. The analytical specificity of the isothermal amplification assay was tested against the target organism SBCMV, the non-target organisms WSSMV and SBWMV, and on healthy wheat plants. The repeatability of the assay was assessed by considering the concordance between the results of each replicate of the same sample obtained under the same conditions, while the reproducibility was defined as the concordance between the results of a single test including aliquots of the same sample analysed under different conditions such as running time, operators and instruments. Three technical replicates for each biological replicate were included in the assay. The best combination of extraction buffer and extract dilution was selected for further analyses.

In some experiments (see below) the reaction mix was added with reverse transcriptase, even though the GspSSD LF DNA Polymerase included in the Isothermal Master Mix ISO-004 has some native reverse-transcriptase activity, to improve the performance of the assay.

#### 2.3.3. Real-Time PCR Primer Design

A primer pair that amplifies a 137 bp DNA fragment, specific for a portion of the CP gene of SBCMV, was designed for real-time PCR (Table 2), based on the following sequences from GenBank: AJ132577, AF146282, KT984978, AF146281, AF146283, AJ252152, AJ298070, FN298362, and AJ298069 (Figure 2). Primers were diluted at 10 μM each.

#### 2.3.4. Real-Time PCR Reaction

The real-time PCR mix consisted of 1 µL of template cDNA, 0.15 µL of each primer, 3.7 µL of nuclease-free water, and 5 µL of 2X iTaq Universal SYBR Green Supermix (Bio-Rad) in a final volume of 10 µL. Thermal cycling conditions were 95 °C for 3 min, followed by 35 cycles of 95 °C for 5 s and 60 °C for 30 s, concluding with 70 °C for 5 min. The cDNA was diluted in sterile water at dilutions from 10^−1^ to 10^−7^. Three technical replicates were carried out for each biological replicate.

#### 2.3.5. Sensitivity Comparison between LAMP and Real-Time PCR Assays

To determine the sensitivity of LAMP and real-time PCR assays for the detection of SBCMV in samples of wheat plants infected through the soil, 10-fold dilutions (10^−3^ to 10^−7^) of the cDNA were used. A cDNA isolated from healthy durum wheat leaves (cv. Claudio) was used as negative control and sterile water was used as a no-template control. 

To perform virus quantification, the coding sequence of SBCMV coat protein was amplified with primers AJ132577_157fw_CP and AJ132577_929rv_CP (Table 2), and the amplicon was cloned into pGEM^®^-T Easy Vector (Promega, Madison, WI, USA), obtaining plasmid pSBCMV_6. The sequence of the insert was determined in both directions and deposited in GenBank under Accession No. OP328757. This plasmid, following linearization, was also used to obtain standard curves (copy number from 10^6^ to 10) and to establish the limit of detection for the two techniques [36,37].

## 3. Results

### 3.1. Development of LAMP Protocol

#### 3.1.1. Choice of Primers and Optimization of Reaction Temperature

Two primer sets (Table 1) designed on the SBCMV CP gene were tested on the cDNA at different dilutions from mechanically infected and healthy wheat samples. Primer set 1 generated several non-specific reactions in cDNAs from negative controls (healthy plants) and was therefore abandoned. Primer set 2 produced positive signals only in cDNAs from SBCMV-infected plants. In temperature gradient tests from 60 to 65 °C, 65 °C was selected for further experiments as the best reaction temperature (Appendix A). Positive samples showed exponential trends from 5 to 10 min and thus LAMP reaction times longer than 20 min were considered unnecessary.

#### 3.1.2. Testing the LAMP Protocol on Different Dilutions of Plant-Derived cDNA and Comparison with Real-Time PCR

Three cDNAs from SBCMV mechanically infected wheat plants and one from a non-infected plant (negative control) were serially diluted from 10^3^ to 10^7^ in sterile water to select a dilution suitable for further tests (Table 3). At all dilutions, except the last one, SBCMV was consistently detected in all three technical replicates. The 10^−5^ dilution of cDNAs was selected for further tests.

The cDNA obtained from 24 wheat leaf samples grown in virus-infected soil were used to compare the sensitivity of LAMP vs. real-time PCR. Each sample was tested in three technical replicates (Table 4). Results indicated that both techniques were able to detect SBCMV in 4 samples (5B, 5C, 171C, and 209A), while all the others were negative.

To determine the limit of detection (LOD) for SBCMV, real-time PCR assays were performed using the pSBCMV_6 plasmid above described, which contains the virus CP sequence. Before real-time PCR, the plasmid was digested with *Apa*I restriction enzyme. The LOD is defined as the lowest quantity at which all technical replicates test positive. Reactions, containing SBCMV 10^6^–10^1^ copy number, were analysed in triplicate by the virus specific real-time PCR assay developed in this study (Figure 3). A R^2^ value of 0.997 was obtained using all the dilutions tested. Ten copies were detected in all technical replicates. Further analyses showed that five copies could not be consistently detected (data not shown), therefore the LOD of ten copies was determined.

The cDNAs from three SBCMV mechanically infected and from one not infected wheat plants were used to determine at which dilution SBCMV can still be reliably detected (Table 5). Dilutions up to 10^−6^ were consistently detected by both techniques, while at 10^−7^ only LAMP produced a reliable result in one sample.

In the first part of the work, where comparisons between the two assays were performed, the CFX96 instrument was used because it allows simultaneous testing of 96 samples. The LAMP protocol was then tested on both CFX and bCUBE instruments, yielding comparable results (data not shown). When LAMP and RT-LAMP assays were done on crude extracts the bCUBE was used, to mimic a possible future use in field conditions using a portable device.

#### 3.1.3. Protocol Specificity

To test the specificity of the LAMP protocol, reactions were performed using cDNA produced from total RNA of plants infected by SBWMV (kindly provided by Dr. Annette Niehl, Julius Kühn-Institut, Braunschweig, Germany) or WSSMV (from the experimental station at the University of Bologna), alongside with SBCMV positive and negative control samples. Amplification signals were observed only in SBCMV-positive control, while no amplification was obtained in samples infected by the two other viruses or in negative control samples (Figure 4). The specificity assay was also performed on total RNA extracts using the RT-LAMP protocol, with similar results (data not shown).

### 3.2. RT-LAMP Protocol Optimisation on Crude Leaf Extracts

#### 3.2.1. Test of Three Different Buffers

Three different extraction buffers were tested for obtaining the crude extracts to be used in LAMP assay. Efficient and specific amplification was obtained in the crude extracts of samples homogenised with the TET and the Tris HCl buffer (Table 6), while the ELISA buffer generated non-specific amplification from the uninfected samples (data not shown). Based on these results, all further experiments were conducted with crude extracts obtained with the TET buffer, because the reaction times were shorter (see Table 6).

#### 3.2.2. Detection of SBCMV in Crude Extracts

The LAMP protocol used in previous tests on cDNAs was tested on crude leaf extracts obtained by the TET buffer from plants grown in infected soil and diluted 10^−4^ in sterile water. The 10^−4^ dilution of crude extracts corresponds to the 10^−5^ dilution of cDNAs in terms of amount of plant material. By calculating the initial amount of leaf material (100 mg) and the final volumes of extracts, we determined that 1 mg of leaf material corresponded to 1 µL of cDNA and to 10 µL of crude extract. According to these calculations, for appropriate comparisons a 10 times higher amount of crude extract had to be used. The plates of bCUBE device can hold 16 samples per plate, so we chose to use one plate for each triplet of samples (each with three technical replicates) including positive, negative, and water control in each plate. Using the cDNAs, four samples tested positive (see Table 4), while using the crude extracts SBCMV was detected only in the sample 209A (Table 7, column RT-LAMP).

#### 3.2.3. Optimization of the RT-LAMP Reaction by Avian Myeloblastosis Virus Reverse Transcriptase

Although the GspSSD LF DNA Polymerase included in the Isothermal Master Mix ISO-004 has some native reverse-transcriptase activity, the performance of the assay can be improved by adding extra reverse transcriptase. To increase the sensitivity of the protocol in detecting virus from crude leaf extracts, 0.5 U of Avian Myeloblastosis Virus Reverse Transcriptase (AMV-RT) (Promega, Madison, WI, USA) was added to each reaction. Only the samples that were found to be positive using cDNAs (see Table 5) were analysed. As above, a reaction plate was prepared for each triplet of samples (each with three technical replicates) including positive, negative, and water control in each plate (Table 7, column RT-LAMP with AMV-RT).

The addition of the AMV-RT allowed the detection of SBCMV from crude extracts also in samples 5B, 5C, and 171C, and anticipated the detection in sample 209A from 10.5 to 6.62 min.

## 4. Discussion

The LAMP assay, following its initial description [24], has been articulated in a number of versions and for different scopes, due to its specificity, sensitivity, robustness, rapidity, and suitability for in-field analyses. Furthermore, it was shown that it does not necessarily require nucleic acid extraction steps. RT-LAMP technique has been used for the detection of several plant viruses with an RNA genome [25]. In particular, specific LAMP and RT-LAMP protocols have been described for the detection of some viruses in wheat [38]. Wheat yellow mosaic virus (WYMV) was also detected from total RNA extracts using a RT-LAMP protocol [39]. Another protocol was developed by Fukuta et al. [35] to detect WYMV, Japanese soil-borne wheat mosaic virus (JSBWMV) and Chinese wheat mosaic virus (CWMV) using the total RNAs and crude extracts, as well. Lee et al. [40] described a procedure to detect wheat streak mosaic virus (WSMV) during quarantine inspections of imported cereals, where LAMP is performed on cDNA following RNA extraction. A LAMP protocol has been proposed also for a DNA virus (i.e., wheat dwarf virus) [41].

In this study, we demonstrated the potential of the RT-LAMP assay for the detection of SBCMV in durum wheat. To the best of our knowledge, no molecular isothermal amplification method has been developed for SBCMV detection in plants using either cDNA products or crude plant extracts.

In order to evaluate the best LAMP primers and amplification conditions, analyses were first carried out on cDNAs obtained from leaves of plants mechanically inoculated with SBCMV. The second step involved the optimisation of the LAMP protocol on crude leaf extracts obtained from durum wheat plants grown in SBCMV-infected soil. 

For LAMP assay, two sets of primers were tested. The first set generated non-specific reactions, while the second set produced signals only in the presence of cDNA from infected plants and was selected for further experiments. A temperature gradient was performed to assess the best reaction temperature, which was found to be 65 °C.

The protocol was tested on different dilutions (10^−3^ to 10^−7^) of cDNA from three mechanically inoculated plants. In all cases, the protocol was able to detect the virus up to the 10^−6^ dilution and in one case at the 10^−7^ dilution (Table 3). A few reports are available on using LAMP for detecting viruses in wheat. WSMV could be detected up to 10^−3^ dilution of cDNA [40]. When RNA extracts were used, WYMV was detected in dilutions up to 10^−5^ [39] or 10^−6^ [35], JSBWMV in dilutions up to 10^−6^, and CWMV up to 10^−4^ [35]. However, it must be considered that results are difficult to compare, since different reagents, protocols, and visualization methods (colorimetric, turbidity, fluorescence) were used.

Due to the uneven distribution of a virus in plant tissues and between plants, we decided to determine how many copies of viral genome could be reliably detected, using serial dilutions of a linearized plasmid containing part of the SBCMV CP gene. Real-time PCR reliably provided positive results down to 10 copies in all technical replicates (Figure 3), which were defined as the detection limit (LOD).

The LAMP performed on cDNAs from infected wheat plants was compared with real-time PCR in terms of sensitivity and both LAMP and real-time PCR successfully detected SBCMV up to the 10^−6^ dilutions (Table 5). We could conclude that LAMP, although being only a qualitative technique, is capable of yielding a positive result in a sample containing down to 10 copies of viral genome. 

When the cDNAs of 24 samples grown in infected soil (not mechanically infected by SBCMV) were analysed using LAMP and real-time PCR assays, both techniques were able to detect SBCMV in four samples (Table 4). This indicates that sensitivity of LAMP is comparable to real-time PCR also when the virus was transmitted by the natural vector in infected soil. It is known that LAMP- and PCR-based assays reach similar levels of sensitivity [38].

The specificity of the SBCMV-LAMP assay was tested on SBWMV (a furovirus closely related to SBCMV) and WSSMV, the other soil-borne transmitted viruses reported on wheat in Europe. No amplification was detected (Figure 4), thus demonstrating the specificity of the SBCMV-LAMP assay developed.

From this first part of the work, we can conclude that the LAMP assay is specific for SBCMV and has the ability to detect the virus in dilutions of cDNA derived from wheat leaf samples at dilutions up to 10^−6^, with a LOD of 10 copies. Most of the protocols developed for detecting plant viruses require nucleic acid purification before performing the assays. Avoiding this step, time from sample collection to results could be reduced, as well as the risks of cross-contamination during manipulations. We therefore tested the technique on crude extracts.

For the optimisation of a RT-LAMP protocol on crude leaf extracts, three extraction buffers were tested. The same LAMP reaction mix used previously for cDNA analysis was employed, because of the Isothermal Master Mix ISO-004 native reverse transcriptase activity. While the ELISA buffer generated non-specific reactions in the healthy samples, the TRIS and the TET buffers showed no visible amplification in those samples. The TET buffer was selected for further experiments because the reaction times were shorter.

When the crude leaf extracts of the 24 samples were analysed by RT-LAMP (Table 7), the addition of AMV-RT enzyme allowed the detection of SBCMV in all four samples that tested positive on cDNA (Table 4). In the single case of sample 209A the virus was detected also without AMV-RT; however, its addition to the reaction mix significantly reduced the reaction time from 10.5 to 6.6 min. The addition of the AMV-RT improves virus detection and therefore should be preferred when crude leaf extracts are tested.

The LAMP assay described in this work proved to be a rapid, simple, specific, and sensitive assay to detect SBCMV in wheat leaves. When working on cDNA, it was found to be as sensitive as real-time PCR. If crude extracts are tested, the RT-LAMP maintained high sensitivity in detecting SBCMV, provided AMV-RT is added. The RT-LAMP assay developed in the present work may pave the way to a wider application of such a procedure in LAMP-based assays of viruses infecting wheat.

## 5. Conclusions

Typically, the molecular analysis of plants for virus detection requires transportation of plant materials from the field to qualified laboratories for RNA extraction procedures and PCR-based tests. These operations are demanding, time-consuming, and expensive, requiring about one working day for accomplishment. For these reasons and due to the lack of in-field SBCMV diagnostic protocols, we set up a rapid RT-LAMP assay for specific and sensitive SBCMV detection that can be completed in less than one hour, including the crude plant extracts preparation. Considering only the cost of consumables for each sample, the real-time PCR approach costs about 6 euros, while the RT-LAMP has a total cost of about 0.8 euro. In future, it will be important to validate the protocol directly in the field using a portable instrument (such as the bCUBE), because this assay represents a potential tool for rapid screening of wheat plant material, useful for extensive investigations of spread and pathogenicity of this virus, which to date remain uncertain.

## Figures and Tables

**Figure 1 viruses-15-00140-f001:**
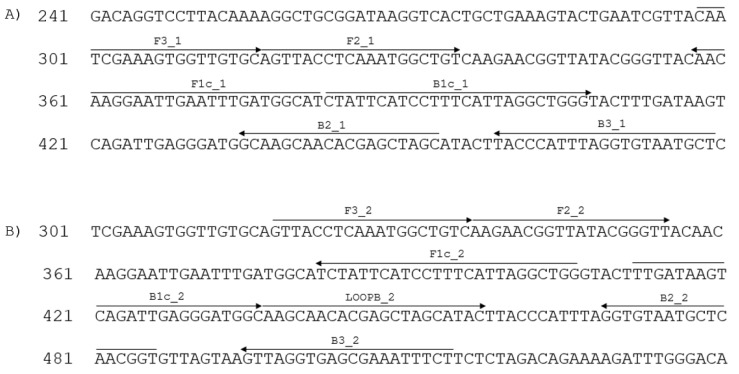
Positions of LAMP primers on SBCMV RNA2 sequence AJ132577. Arrowheads indicate the orientation of the primer. (**A**): primer set 1; (**B**): primer set 2.

**Figure 2 viruses-15-00140-f002:**
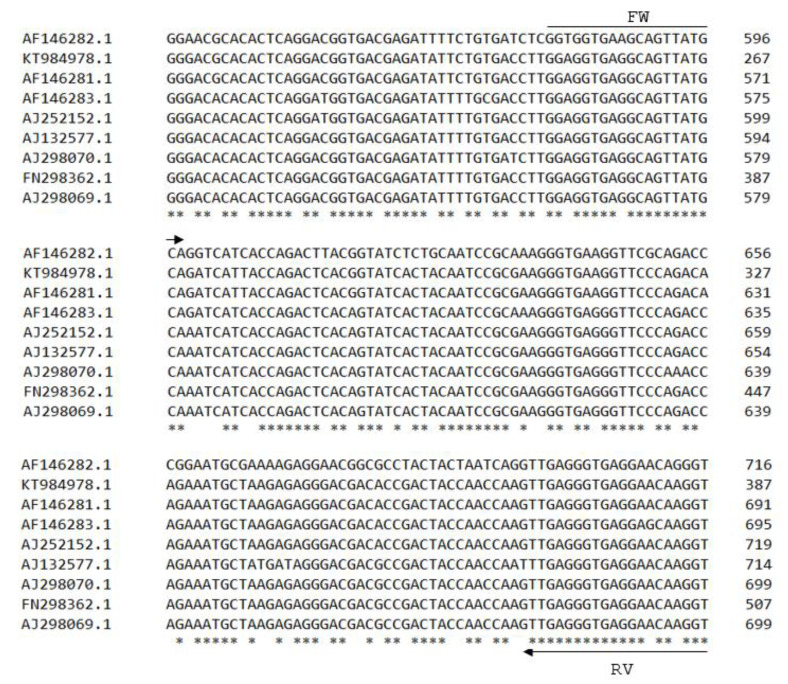
Sequence alignment of a portion of SBCMV RNA 2 of nine SBCMV isolates available in GenBank. Arrows indicate the position of primers designed for real-time PCR.

**Figure 3 viruses-15-00140-f003:**
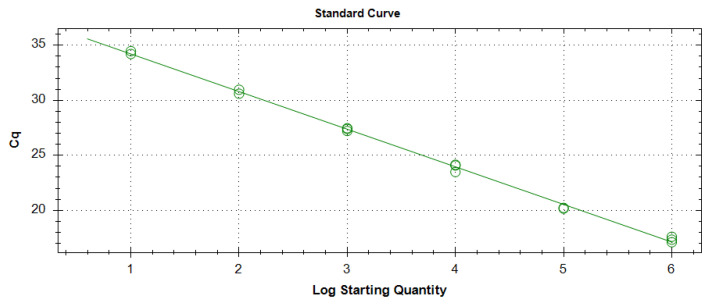
Standard curve obtained using 10-fold serial dilutions of linearized pSBCMV_6 plasmid. Experiment performed in the CFX instrument.

**Figure 4 viruses-15-00140-f004:**
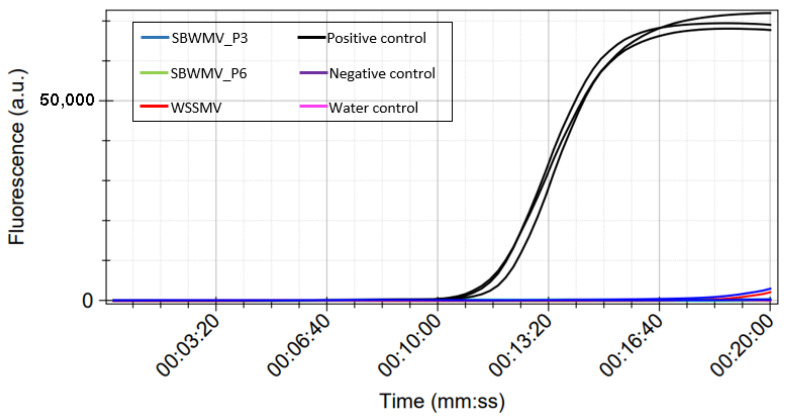
LAMP test performed on cDNA from samples infected with SBWMV and WSSMV diluted 10^−4^. Experiment performed in the bCUBE instrument.

**Table 1 viruses-15-00140-t001:** LAMP primers for detection of SBCMV. Genome position (expressed in nt) refer to GenBank sequence AJ132577. Primers F1c and F2 together build the FIP primer, whereas primers B1c and B2 together build the BIP primer of the different LAMP primer sets.

Primer Set	Name	Sequence (5′–3′)	Genome Position	Concentration Used in LAMP Reaction
1	F3_1	CAATCGAAAGTGGTTGTGC	298–316	0.25 μM
	B3_1	AGCATTACACCTAAATGGGTA	459–479	0.25 μM
	FIP_1	ATGCCATCAAATTCAATTCCTTGTT-AGTTACCTCAAATGGCTGT	358–382 (F1c_1) + 317–335 (F2_1)	2.5 μM
	BIP_1	CTATTCATCCTTTCATTAGGCTGGG-GCTAGCTCGTGTTGCTTG	383–417 (B1c_1) + 436–453 (B2_1)	2.5 μM
2	F3_2	GTTACCTCAAATGGCTGTC	318–336	0.25 μM
	B3_2	AGAAATTTCGCTCACCTAAC	495–514	0.25 μM
	FIP_2	CCAGCCTAATGAAAGGATGAATAGA-AAGAACGGTTATACGGGTT	382–406 (F1c_2) + 337–355 (F2_2)	2.5 μM
	BIP_2	TTGATAAGTCAGATTGAGGGATGGC-ACCGTTGAGCATTACACC	412–436 (B1c_2) + 469–486 (B2_2)	2.5 μM
	LOOPB_2	AAGCAACACGAGCTAGCATACTTAC	437–461	1.25 μM

**Table 2 viruses-15-00140-t002:** Degenerate real-time PCR primers for detection of SBCMV and primers for amplification of the complete coat protein gene.

Primer Name	Sequence (5′–3′)	Genome Position	Fragment Size (nt)
qPCR_577fw	GGWGGTGARGCAGTTATGC	577–595	137
qPCR_714rv	ACCYTGYTCCTCACCCTCAA	695–714	
AJ132577_157fw_CP	GGTAGTCAGCTGTTAGCGTGT	157–177	773
AJ132577_929rv_CP	TCGGCCAAAACCAGCCTATT	910–929	

**Table 3 viruses-15-00140-t003:** LAMP assay on serial dilutions of cDNA from three SBCMV-infected wheat plants mechanically inoculated. Experiment performed in the CFX instrument.

Sample	cDNA Dilution	LAMP
Positive 1	10^−3^	+/+/+
	10^−4^	+/+/+
	10^−5^	+/+/+
	10^−6^	+/+/+
	10^−7^	+/−/−
Positive 2	10^−3^	+/+/+
	10^−4^	+/+/+
	10^−5^	+/+/+
	10^−6^	+/+/+
	10^−7^	+/+/+
Positive 3	10^−3^	+/+/+
	10^−4^	+/+/+
	10^−5^	+/+/+
	10^−6^	+/+/+
	10^−7^	+/+/−
Negative	10^−3^	−/−/−
	10^−4^	−/−/−
	10^−5^	−/−/−
	10^−6^	−/−/−
	10^−7^	−/−/−

**Table 4 viruses-15-00140-t004:** Comparison between LAMP and real-time PCR performed on cDNAs (10^−5^ dilution) of leaf samples from durum wheat grown in virus-infected soil. Experiment performed in the CFX instrument. Rt: reaction time; Tm: melting temperature; Cq: threshold cycle; SD: standard deviation; nd: not detected.

Sample	LAMP		Real-Time PCR	
(cDNA)	Rt (min) ± SD	Tm (°C) ± SD	Cq ± SD	Tm (°C) ± SD
1_A	nd	nd	nd	nd
1_B	nd	nd	nd	nd
1_C	nd	nd	nd	nd
5_A	nd	nd	nd	nd
5_B	8.6 ± 0.7	84.5 ± 0.0	30.06 ± 0.26	80.5 ± 0.0
5_C	6.13 ± 0.06	84.5 ± 0.0	22.8 ± 0.3	80.17 ± 0.29
166_A	nd	nd	nd	nd
166_B	nd	nd	nd	nd
166_C	nd	nd	nd	nd
171_A	nd	nd	nd	nd
171_B	nd	nd	nd	nd
171_C	8.25 ± 0.10	84.5 ± 0.0	29.55 ± 0.27	80.5 ± 0.0
209_A	6.81 ± 0.13	84.5 ± 0.0	25.46 ± 0.26	80.5 ± 0.0
209_B	nd	nd	nd	nd
209_C	nd	nd	nd	nd
210_A	nd	nd	nd	nd
210_B	nd	nd	nd	nd
210_C	nd	nd	nd	nd
213_A	nd	nd	nd	nd
213_B	nd	nd	nd	nd
213_C	nd	nd	nd	nd
214_A	nd	nd	nd	nd
214_B	nd	nd	nd	nd
214_C	nd	nd	nd	nd
Positive control	9.40 ± 0.22	84.5 ± 0.0	18.238 ± 0.026	80.17 ± 0.29
Negative control	nd	nd	nd	nd

**Table 5 viruses-15-00140-t005:** Comparison between LAMP and real-time PCR on serial dilutions of cDNA from three wheat plants mechanically inoculated with SBCMV. Experiment performed in the CFX instrument. Rt: reaction time; Cq: threshold cycle; EC: estimated copy number; SD: standard deviation; nd: not detected.

Sample	Diluition	LAMP Tp (min) ± SD	qPCR Cq ± SD	qPCR Quantification EC ± SD
Positive 1	10^−3^	5.327 ± 0.029	22.70 ± 0.15	24,700 ± 2500
	10^−4^	6.28 ± 0.18	26.02 ± 0.13	2720 ± 220
	10^−5^	7.77 ± 0.19	(29.71 ± 0.21) **	240 ± 30
	10^−6^	9.2 ± 0.9	(32.13 ± 0.06) **	47.2 ± 1.8
	10^−7^	nd *	nd	nd
Positive 2	10^−3^	4.90 ± 0.15	21.357 ± 0. 017	60,000 ± 700
	10^−4^	5.60 ± 0.06	24.277 ± 0.025	8650 ± 140
	10^−5^	7.08 ± 0.19	27.5 ± 0.4	1120 ± 240
	10^−6^	8.6 ± 1.3	29.97 ± 0.11	199 ± 14
	10^−7^	9.6 ± 1.4	nd	nd
Positive 3	10^−3^	4.99 ± 0.16	21.59 ± 0.09	51,000 ± 3000
	10^−4^	5.84 ± 0.05	24.9 ± 0.4	5900 ± 1400
	10^−5^	6.87 ± 0.17	27.90 ± 0.16	780 ± 90
	10^−6^	9.6 ± 0.6	30.6 ± 0.4	130 ± 30
	10^−7^	(9.3 ± 0.4) **	nd	nd
Negative	10^−3^	nd	nd	nd
	10^−4^	nb	nb	nd
	10^−5^	nb	nb	nd
	10^−6^	nb	nb	nd
	10^−7^	nb	nb	nd

* 1 positive and 2 negatives over 3 technical replicates. ** 2 positives and 1 negative over 3 technical replicates.

**Table 6 viruses-15-00140-t006:** Comparison of RT-LAMP assay on crude extracts prepared from mechanically inoculated plants at different dilutions in TET and Tris buffers. Experiment performed in the bCUBE instrument. Rt: reaction time; SD: standard deviation; nd: not detected.

Sample	Dilution	TET Rt (min) ± SD	Tris HCl Rt (min) ± SD
	1:10^3^	11.4 ± 0.7	12.4 ± 0.5
**Positive control**	1:10^4^	13.00 ± 0.29	13.6 ± 1.5
	1:10^5^	13.4 ± 0.8	15.2 ± 0.8
	1:10^3^	nd	nd
**Negative control**	1:10^4^	nd	nd
	1:10^5^	nd	nd
**Water control**	-	nd	nd

**Table 7 viruses-15-00140-t007:** Detection of SBCMV using RT-LAMP mix and RT-LAMP mix containing AMV-RT in leaf crude extracts (dilution 10^−4^) of 24 samples from durum wheat plants grown in SBCMV-infected soil. Experiment was performed in the bCUBE instrument. Rt: reaction time; SD: standard deviation; nd: not detected; nt: not tested.

Sample (Crude Extract)	RT-LAMP Rt (min) ± SD	RT-LAMP with AMV-RT Rt (min) ± SD
1A	nd	nt
1B	nd	nt
1C	nd	nt
5A	nd	nd
5B	nd	6.99 ± 0.10
5C	nd	6.49 ± 0.08
166A	nd	nt
166B	nd	nt
166C	nd	nt
171A	nd	nd
171B	nd	nd
171C	nd	7.43 ± 0.23
209A	10.5 ± 0.8	6.62 ± 0.03
209B	nd	nd
209C	nd	nd
210A	nd	nd
210B	nd	nd
210C	nd	nd
213A	nd	nt
213B	nd	nt
213C	nd	nt
214A	nd	nt
214B	nd	nt
214C	nd	nt
Positive	(11.6 ± 1.2) *	(6.1 ± 0.6) **
Negative	nd	nd
Water	nd	nd

* Mean of the technical replicates in eight plates. ** Mean of the technical replicates in three plates.

## Data Availability

Data is contained within the article or Appendix A.

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
