# Peer review of "Fast and Sensitive Detection of Soil-Borne Cereal Mosaic Virus in Leaf Crude Extract of Durum Wheat"

_viruses, 2022, doi:10.3390/v15010140_

Round 1

Reviewer 1 Report

Dear Elisabeth Yi,

Assistant Editor, Viruses

I have read and revised the manuscript viruses-2091114 entitled "Fast and sensitive detection of soil-borne cereal mosaic virus in leaf crude extract of durum wheat ". Soil-borne cereal mosaic virus causes a serious viral disease which can reduce wheat yield up to 70%. SBCMV is transmitted by the soil-borne plasmodiophorid – Polymyxa graminis. Viruliferous resting spores and zoospores of its vector can survive in the soil for decades and can be dangerous for susceptible cereals. For that reason rapid and sensitive diagnostic method is necessary to assess the actual spread of the virus.

            The manuscript presents molecular diagnostics of SBCMV based on RT LAMP technique which is convenient tool frequently used in the diagnostic of various pathogens. The development of new diagnostic methods, that can be performed under field conditions, are worth to be published however, many parts of this manuscript are unclear or include methodical mistakes and editorial errors (see additional comments listed below). Moreover, the manuscript requires revision in English language. Take all of these into account, in my opinion, this paper can’t be accepted to Viruses in current form. I strongly encourage the Authors to improve the quality of the paper and then resubmit.

General comments

1) The major part of the manuscript concerns testing of the optimization and sensitivity of the LAMP reaction, while the main goal of presented study is detection of SBCMV particles directly in crude sap of infected plants, using RNA and not cDNA, in RT-LAMP. I recommend changing these proportions in the text and paying more attention to the description of the results for RT-LAMP reaction.

2) In general, there is a lot of confusion in the use of the terms LAMP and RT-LAMP. It seems as if the authors did not distinguish between these two reactions.  

Example:

Line 212: "3.1. Development of RT-LAMP protocol"

This whole part concerns LAMP not RT-LAMP reaction, because described technique is based on cDNA and no reverse transcriptase was used!

Line 286: header -„RT-LAMP protocol….", while section concerning RT-LAMP is placed only in lines 311-340.

Lines 390-390: SBCMV was detected in 4 samples by real-time PCR and RT-LAMP not LAMP, because in LAMP reaction without AMV-RT virus presence was confirmed in only one sample (209A). Thus, this description concerns sensitivity of RT-LAMP.

Futhermore, it is not clear whether a specificity assay was performer also for RT-LAMP as for LAMP (lines 395-398). I couldn't find any information about it in the text.

3) LAMP and RT-LAMP are qualitative not quantitative diagnostic tools. In my opinion, to assess the sensitivity of the developed technique, the use of 10-fold dilutions of total RNA in the RT-LAMP reaction and real-time RT-PCR would be more appropriate. Such solutions were used in the studies cited in this manuscript [References:  35, 36 and 37] as well as in Ratti et al. 2004, where sensitivity comparison between TaqMan PCR, conventional RT/PCR and ELISA assays was estimated. Moreover, using of serial dilution of pSBCMV_6 plasmid to establish the limit of detection is methodical incorrect and gave unreal results. I recommend improving this part of studies.

4) The use of an additional reverse transcriptase in RT-LAMP have been frequently described in published protocols, also cited in this manuscript. Therefore, whole fragment (lines 311-322) should be placed much earlier into Material and Methods (lines 166-169) and not only in Results (lines 313-322).  

5) In this study two sets of primers were designed:

- For real-time PCR the primers based on alignment of available in GenBank SBCMV sequences.

 - For LAMP reaction based only on one of them.

Why the authors did not compare these sequences and determine the most conserved regions to design the primers for the LAMP reaction?

6) In Table 1. the authors placed LOOPB_2 (Backward Loop primer) without Forward Loop Primer and only in set 2 . Is this correct?

7) I recommend being as precise as possible:

A) Line 93: Characteristics of Italian SBCMV isolate should appear at the beginning of Material and Methods section (e.g. Accession numer from GenBank) while such information (OP328757) appears in lines 202-203.  This SBCMV isolate came from „an experimental station at the University of Bologna”. It has been probably used in the experiments described earlier.

B) Line 100: I propose add a development phase on the BBCH scale. It will be more precise.

C) In this research  two types of plant material were used: mechanically inoculated and  grown in SBCMV-infected soil. The description of the results is sometimes ambiguous and the description of which material they refer to is included only in the table header.  This is not sufficient.

D) Lines 106-110: What method was used to confirm SBCMV infection before total RNA extraction?

E) Lines 108-110: There is no information which primer was used for cDNA synthesis. Specific reverse primer or random-hexamers?

F) Line 229: In results: „the 10-5 dilution of cDNA was selected for further tests” while for testing specificity of the LAMP protocol SBWMV and WSSMV cDNAs „diluted  10-4” were used (line 284). Why?

G) Lines 305-306: „The 10-4 dilution of crude extracts corresponds to the 10-5 dilution of cDNA in terms of amount of plant material”.  Where are results that prove this.

H) The authors must standardize the notation of the dilutions used:

Line 120: (10-3 to 10-5),

Line 190: from 1: 10 to 1:107

Line 226: from 103 to 107

Line 19: common wheat , line 46: bread wheat 

Editorial errors (some examples)

Lanes 22 and 84: I propose to use dispersal instead diffusion [dispersal -the action or process of distributing things or people over a wide area].

Lanes 25-26: RT-LAMP protocol should be compare with real-time RT-PCR instead real-time PCR.

Line 119: Sentence starts with the numer 100.

Line 133: I suggest replacing „coordinates” with „genome position” and „bp” with „nt”

Coordinates appear also in table 2.

Lines 350, 353: I propose replace „Fukuta and coworkers” to „Fukuta et al.” and „Lee and coworkers” to „Lee et al.”

Reviewer 2 Report

The paper describes different parameters of a newly developed RT-LAMP test to detect soil-borne cereal mosaic virus. Two different primer sets were compared, as well as different types of sample preparation, which is very useful. The sensitivity of the RT-LAMP test is compared with that of real-time RT-PCR and the specificity is established using similar viruses SBWMV and WSSMV, known to infect wheat. The language of the paper is excellent. I suggest writing "along the veins" instead of "along to the veins" on line 52. And add the terms "RT-LAMP" to the words "Temperature gradient tests", and SBCMV to "cDNA", in the description of Figure S1.

Author Response

We thank the reviewer for appreciating our work and for providing useful comments.

The suggestions have been incorporated in the revised text.

Round 2

Reviewer 2 Report

I suggest accepting the present form